# Testing for Local Spatial Association Based on Geographically Weighted Interpolation of Geostatistical Data with Application to PM2.5 Concentration Analysis

**Fen-Jiao Wang [1], Chang-Lin Mei [1,*], Zhi Zhang [2] and Qiu-Xia Xu [1]**

[1]   Department of Finance and Statistics, School of Science, Xi'an Polytechnic University, Xi'an 710048, China
[2]   Department of Statistics, School of Mathematics and Statistics, Xi'an Jiaotong University, Xi'an 710049, China
*   Correspondence: clmei@xpu.edu.cn

**Abstract:** Using local spatial statistics to explore local spatial association of geo-referenced data has attracted much attention. As is known, a local statistic is formulated at a particular sampling unit based on a prespecific proximity relationship and the observations in the neighborhood of this sampling unit. However, geostatistical data such as meteorological data and air pollution data are generally collected from meteorological or monitoring stations which are usually sparsely located or highly clustered over space. For such data, a local spatial statistic formulated at an isolate sampling point may be ineffective because of its distant neighbors, or the statistic is undefinable in the sub-regions where no observations are available, which limits the comprehensive exploration of local spatial association over the whole studied region. In order to overcome the predicament, a local-linear geographically weighted interpolation method is proposed in this paper to obtain the predictors of the underlying spatial process on a lattice spatial tessellation, on which a local spatial statistic can be well formulated at each interpolation point. Furthermore, the bootstrap test is suggested to identify the locations where local spatial association is significant using the interpolated-value-based local spatial statistics. Simulation with comparison to some existing interpolation and test methods is conducted to assess the performance of the proposed interpolation and the suggested test methods and a case study based on PM2.5 concentration data in Guangdong province, China, is used to demonstrate their applicability. The results show that the proposed interpolation method performs accurately in retrieving an underlying spatial process and the bootstrap test with the interpolated-value-based local statistics is powerful in identifying local patterns of spatial association.

**Keywords:** geographically weighted interpolation; local spatial statistic; local spatial association; bootstrap

## 1. Introduction

Spatial autocorrelation or association is one of the fundamental properties of spatial data [1] and the exploration of spatial association patterns is of great importance in understanding the intrinsic characteristics of underlying processes and making related decisions in applications. However, due to the other fundamental property of spatial heterogeneity or non-stationarity of spatial data, the assumption of stationarity or structural stability over space may be highly unrealistic [2]. Therefore, more emphasis has been placed on the development of local modeling methodologies for spatial data analysis [3,4]. In the analysis of spatial association in particular, the use of local spatial statistics to explore local patterns of spatial association has attracted considerable attention in recent decades (for general overviews, see for example [5–7]). Among many kinds of local spatial statistics, the most popular ones are perhaps Getis and Ord's $G_i$ and $G_i^*$ [8,9] and Anselin's LISAs [2] including the most commonly used local Moran's $I_i$ and Geary's $c_i$, which mainly focus on exploring local clusters of similarity (either large values or small values) or dissimilarity among spatial data, a very important kind of spatial patterns in practical applications. As

a class of the most used local statistics, both LISAs and $G_i$ or $G_i^*$ have been extended to spatiotemporal data for exploring spatiotemporal association [10–12]. In addition, the $c_i$ statistic has also been extended in the multivariate setting for measuring multivariate spatial autocorrelation [13]. Driven by substantial practical problems related to the exploration of spatial association patterns, a lot of local spatial or spatiotemporal statistics have been proposed and applied to a variety of practical fields. One can refer to the references [14–16] for an overview of local spatial statistics with their extensive applications.

In view of the fact that the observations of a variable are collected generally with noise such as measurement error and/or the influence of some uncontrollable factors, it is reasonable to treat a local spatial statistic as a random variable. Therefore, statistical significance tests are essential for a local spatial statistic to be used to detect spatial association patterns in order to make the analyzing results have a solid statistical basis. In such tests, the null or reference distribution (i.e., the distribution of a local statistic under the null hypothesis that no spatial association exist at the focal location) plays a fundamental role in deriving *p*-value of the test. Several methodologies for the derivation of the null distributions of local spatial statistics have been developed. The commonly used methods include the asymptotic, especially normal distribution approximation [2,9,17,18], the approximation based on the distribution theory of quadratic forms in normal variables [19,20], and the Monte Carlo approximation including the randomized permutation and the bootstrap methods [2,12,21,22]. In particular, Mei et al. [23] recently proved that the bootstrap approximation to the null distributions of the local Moran's $I_i$, local Geary's $c_i$, and local Getis and Ord's $G_i^*$ are statistically consistent, which establishes a theoretical basis for the bootstrap approximation to the null distributions of these commonly used local spatial statistics.

A local spatial statistic is a measure of spatial association at a particular spatial unit where the observation of the interested variable is collected. It is constructed based on the spatial proximity relationship characterized by the pre-specified weights and the observations in the neighborhood of this particular spatial unit. According to Cressie [24], spatial data can be mainly categorized as regional data observed at a finite collection of locations or regular/irregular areas and as geostatistical data collected from a continuous surface defined on the studied geographical region. For regional data and such geostatistical data that are collected at relatively densely and uniformly distributed locations over the region, a local spatial statistic can be well formulated based on the observations and the proximity relationship among the areas or locations with, for example, the rook's or queen's tessellation or the *k*-nearest neighborhood scheme. However, many real-life geostatistical data sets such as meteorological data and air pollution data are only available at the observation stations which are usually unevenly located over space, leading to very sparse or isolate sampling locations in some areas and dense or highly clustered ones in other areas. For example, the left panel of Figure 2 in Section 2 shows the 101 air quality monitoring stations in Guangdong province, China, from which we can observe that the stations are highly clustered in the center part of southeast area and very sparse in the north and west areas with several counties having only one or two stations. For such geostatistical data set, although one can routinely formulate a local spatial statistic of spatial association at an isolate sampling location, the statistic may be ineffective in exploring local spatial association because its distant neighbors violate the Tobler's first law of geography that near things are more correlated than distant things [25]. Moreover, a local spatial statistic could not be formulated in the extensive west boundary area where there are no monitoring stations, which limits the comprehensive exploration of local spatial association patterns of an underlying process over the whole province.

Nevertheless, for geostatistical data, as the underlying process from which the geostatistical data are collected is assumed to be continuous, interpolation at, for example, a lattice spatial tessellation seems a promising methodology to deal with the above predicament. Based on the interpolated values at the lattice spatial tessellation, a local spatial statistic can be formulated for the effective and comprehensive exploration of local spatial

association patterns of the underlying process, provided that the interpolation method can yield accurate interpolated values at the latticed locations.

Among many interpolation methods, kriging has been one of the most popular interpolation approaches and many types of kriging have been developed for different orientations [24]. Although the standard kriging can obtain the best linear unbiased predictor of an underlying spatial process at any an unsampled location, it needs the assumption that the covariance function of the process is fully known, which is usually unobtainable in practice. To handle this problem, a spatial process which the kriging interpolation focus on is assumed to be second-order stationary and a pre-specified parametric homogeneous semi-variogram is fitted by the observations to yield the interpolated values. The homogeneous semi-variogram may violate the fundamental heterogeneity property of spatial data and, as a result, the kriging interpolation may fail in yielding accurate predictors.

Another most commonly used interpolation method for spatial data is perhaps the inverse distance weighted (IDW) interpolation [26], in which the interpolation value at a given location is a weighted average of the observations collected at sampling locations and the weight for each observation is inverse of the distance between the interpolation location and the corresponding sampling location to some positive power with the power parameter generally taken to be 1 or 2. The rationale of IDW interpolation is in line with the Tobler's first law of geography [25] and IDW is in fact a non-parametric smoothing technique with fixed weights for a given interpolation location. Because of the fixed weights for a given interpolation location, the IDW interpolation method may also produce less accurate interpolated values even for a continuous spatial process.

Although the IDW interpolation method is not flexible enough to adaptively yield interpolated values, this non-parametric methodology motivates us to improve interpolation method by using the local smoothing techniques [27]. In the recent decades, local smoothing techniques have been well established in the literature of non-parametric regression, which can adaptively fit heterogeneity of the underlying process by embedding an unfixed smoothing parameter or bandwidth into the weights with the optimal bandwidth size selected by some data driven criterion, making them especially suitable for the interpolation of a continuous surface. In particular, the local polynomial smoother has been theoretically shown having good asymptotic behaviors in bias and variance of the predictor and the attractive property of automatically correcting the boundary effect, i.e., resulting in an equally accurate predictor on the boundary and in the inner of the domain of the underlying process [27]. As the most commonly used local polynomial smoother, the local-linear smoothing methodology is extended in this article to derive the interpolated values for a spatial process. Because the extended local-linear smoother for a spatial process is in fact a locally weighted least-squares procedure with spatial distance decay weights, we call the interpolation method the local-linear geographically weighted interpolation (LGWI) henceforth. Based on the interpolated values on a lattice tessellation, any a local spatial statistic such as the local Moran's $I_i$ or the Getis and Ord's $G_i^*$ can be formulated to measure local spatial association among the observations of an underlying spatial process.

As aforementioned, formal statistical tests are another important issue related to the exploration of spatial association using local spatial statistics and several kinds of methods have been developed. Nevertheless, the normal distribution approximation has been empirically or theoretically shown to be sometimes problematic [2,18,28,29]. The approximation based on the quadratic form distribution theory is closely related to the assumption that the data is normally distributed [19,20], which might be invalid to many real-life data sets. Fueled by modern computers, the randomized permutation and the bootstrap approximations have become an attractive way to derive the null distributions of the local spatial statistics [2,8,12,22]. Especially, in view of the theoretical result that the bootstrap approximation to the null distributions of the commonly used local spatial statistics is consistent [23] and the bootstrap method is in fact free of the assumption that the observations are collected from a normal distribution, we therefore suggest the bootstrap

approximation to derive the *p*-values of the related tests for identifying the locations where local spatial association is statistically significant.

The rest of this article is organized as follows. In Section 2, the LGWI method and the interpolated-value-based local spatial statistics with the bootstrap test for the significance of local spatial association are introduced, respectively; the synthetic data from a properly designed experiment and the real-life data of PM2.5 concentration in Guangdong province, China, are then formulated to evaluate the performance of both the interpolation and the test methods and demonstrate their applicability, respectively. The related analyzing results with the comparison to some existing interpolation and test methods are reported in Section 3. The paper is ended with conclusion and discussion.

## 2. Methods and Data Sources for Evaluation of the Methods and Demonstration of Their Applicability

*2.1. Local-Linear Geographically Weighted Interpolation and Interpolated-Value-Based Local Spatial Statistics with the Bootstrap Test for Significance of Local Spatial Association*

2.1.1. Local-Linear Geographically Weighted Interpolation (LGWI)

Let $Y$ be the interested attribute variable and let $f(u, v)$, an unknown function of the spatial coordinates $(u, v)$ defined on the region $D$, be the underlying spatial process from which the observations of $Y$ are collected. Considering noise, we formulate the following non-parametric regression model between $Y$ and $f(u, v)$:

$$Y = f(u, v) + \varepsilon,$$

where $\varepsilon$ is the random error term with $\mathrm{E}(\varepsilon) = 0$ and $\mathrm{Var}(\varepsilon) = \sigma^2 > 0$. Let $\{y_i\}_{i=1}^n$ be the observations of $Y$ collected at the spatial locations or sampling points $\{(u_i, v_i)\}_{i=1}^n$. The resulting sample form of the above model shows

$$y_i = f(u_i, v_i) + \varepsilon_i, \quad i = 1, 2, \cdots, n. \tag{1}$$

Suppose further that $f(u, v)$ is of continuous partial derivatives with respect to $u$ and $v$, respectively, which we denote by $f^{(u)}(u, v)$ and $f^{(v)}(u, v)$ henceforth. Given an interpolation point $(u_0, v_0)$ and according to the Taylor's expansion, $f(u, v)$ can be approximated in the neighborhood of $(u_0, v_0)$ by

$$f(u, v) \approx f(u_0, v_0) + f^{(u)}(u_0, v_0)(u - u_0) + f^{(v)}(u_0, v_0)(v - v_0).$$

The LGWI predictor of $f(u, v)$ at $(u_0, v_0)$ is the solution of $f(u_0, v_0)$ in the following locally weighted least-squares problem. Namely, minimize the objective function

$$\sum_{i=1}^n \left( y_i - f(u_0, v_0) - f^{(u)}(u_0, v_0)(u_i - u_0) - f^{(v)}(u_0, v_0)(v_i - v_0) \right)^2 w_i(u_0, v_0) \tag{2}$$

with respect to $f(u_0, v_0)$, $f^{(u)}(u_0, v_0)$ and $f^{(v)}(u_0, v_0)$, where $\{w_i(u_0, v_0)\}_{i=1}^n$ are the weights at $(u_0, v_0)$. Let

$$\boldsymbol{X}(u_0, v_0) = \begin{pmatrix} 1 & u_1 - u_0 & v_1 - v_0 \\ 1 & u_2 - u_0 & v_2 - v_0 \\ \vdots & \vdots & \vdots \\ 1 & u_n - u_0 & v_n - v_0 \end{pmatrix}, \ \boldsymbol{y} = \begin{pmatrix} y_1 \\ y_2 \\ \vdots \\ y_n \end{pmatrix}, \ \boldsymbol{a}(u_0, u_0) = \begin{pmatrix} f(u_0, v_0) \\ f^{(u)}(u_0, v_0) \\ f^{(v)}(u_0, v_0) \end{pmatrix},$$

and

$$\boldsymbol{W}(u_0, v_0) = \mathrm{Diag}(w_1(u_0, v_0), w_2(u_0, v_0), \cdots, w_n(u_0, v_0)). \tag{3}$$

Then the solution of the above optimization problem is

$$
\hat{\boldsymbol{a}}(u_0, v_0) = \left( \hat{f}(u_0, v_0), \hat{f}^{(u)}(u_0, v_0), \hat{f}^{(v)}(u_0, v_0) \right)^{\mathrm{T}}
$$
$$
= \left( \boldsymbol{X}^{\mathrm{T}}(u_0, v_0) \boldsymbol{W}(u_0, v_0) \boldsymbol{X}(u_0, v_0) \right)^{-1} \boldsymbol{X}^{\mathrm{T}}(u_0, v_0) \boldsymbol{W}(u_0, v_0) \boldsymbol{y}
$$

(4)

and the LGWI predictor of $f(u, v)$ at $(u_0, v_0)$, which we denote by $z_0$, is then

$$
z_0 = \hat{f}(u_0, v_0) = (1, 0, 0) \hat{\boldsymbol{a}}(u_0, v_0) = (1, 0, 0) \boldsymbol{Q}(u_0, v_0) \boldsymbol{y},
$$

(5)

where

$$
\boldsymbol{Q}(u_0, v_0) = \left( \boldsymbol{X}^{\mathrm{T}}(u_0, v_0) \boldsymbol{W}(u_0, v_0) \boldsymbol{X}(u_0, v_0) \right)^{-1} \boldsymbol{X}^{\mathrm{T}}(u_0, v_0) \boldsymbol{W}(u_0, v_0).
$$

(6)

The weights $\{w_i(u_0, v_0)\}_{i=1}^{n}$ at $(u_0, v_0)$ are generated by a kernel function, usually the Gaussian or bisquare kernel with a fixed or adaptive bandwidth [30]. For irregular sampling points, Gollini et al. [31] recommended the use of the weights with an adaptive bandwidth to perform the locally smoothing procedure. Specifically, given an integer $k$, let $d_{0k}$ be the Euclidean distance from $(u_0, v_0)$ to its $k$-th nearest sampling point and $\{d_{0i}\}_{i=1}^{n}$ be the Euclidean distances from $(u_0, v_0)$ to all of the sampling points $\{(u_i, v_i)\}_{i=1}^{n}$. Then the weights with an adaptive bandwidth are generated by

$$
w_{i(k)}(u_0, v_0) = \begin{cases} \left( 1 - \left( \frac{d_{0i}}{d_{0k}} \right)^2 \right)^2, & \text{if } d_{0i} \leq d_{0k}; \\ 0, & \text{otherwise,} \end{cases} \quad i = 1, 2, \cdots, n,
$$

(7)

where $d_{0k}$ is the adaptive bandwidth, which is in general different for a different interpolation point $(u_0, v_0)$ and guarantees that $k$ observations of $Y$ in the neighborhood of $(u_0, v_0)$ are used to derive the interpolated value $z_0$ at $(u_0, v_0)$.

The parameter $k$, which is a proxy of the adaptive bandwidth and will be called pseudo-bandwidth henceforth, plays a key role in the local smoothing technique and its optimal size should be firstly determined based on the available data $\{y_i; (u_i, v_i)\}_{i=1}^{n}$ in order to implement the foregoing interpolation procedure at any an unsampled location. Here, the AICc criterion [30] is used to search for the optimal size of $k$. Specifically, given an integer $k$, set $(u_0, v_0)$ in Equation (5) to be each of the sampling points $\{(u_i, v_i)\}_{i=1}^{n}$ and compute the fitted value of $Y$ at $(u_i, v_i)$, which we denote by $\hat{y}_i(k)$, yielding

$$
\hat{y}_i(k) = \hat{f}(u_i, v_i) = (1, 0, 0) \boldsymbol{Q}(u_i, v_i) \boldsymbol{y}, \quad i = 1, 2, \cdots, n.
$$

Then the fitted vector of $Y$ at $n$ sampling points can be expressed as

$$
\hat{\boldsymbol{y}} = (\hat{y}_1(k), \hat{y}_2(k), \cdots, \hat{y}_n(k))^{\mathrm{T}} = \boldsymbol{H}(k) \boldsymbol{y},
$$

(8)

where

$$
\boldsymbol{H}(k) = \begin{pmatrix} (1, 0, 0) \boldsymbol{Q}(u_1, v_1) \\ (1, 0, 0) \boldsymbol{Q}(u_2, v_2) \\ \vdots \\ (1, 0, 0) \boldsymbol{Q}(u_n, v_n) \end{pmatrix}
$$

is the hat matrix of the local-linear smoother. The residual sum of squares is then

$$
\mathrm{RSS}(k) = (\boldsymbol{y} - \hat{\boldsymbol{y}})^{\mathrm{T}} (\boldsymbol{y} - \hat{\boldsymbol{y}}) = \boldsymbol{y}^{\mathrm{T}} (\boldsymbol{I} - \boldsymbol{H}(k))^{\mathrm{T}} (\boldsymbol{I} - \boldsymbol{H}(k)) \boldsymbol{y},
$$

and the AICc score is

$$
\mathrm{AICc}(k) = \log \left( \frac{1}{n} \mathrm{RSS}(k) \right) + \frac{n + \mathrm{tr}(\boldsymbol{H}(k))}{n - 2 - \mathrm{tr}(\boldsymbol{H}(k))},
$$

(9)

where $\mathrm{tr}(\boldsymbol{H}(k))$ stands for the trace of $\boldsymbol{H}(k)$. The optimal size of the pseudo-bandwidth $k$ is such the integer that minimizes $\mathrm{AICc}(k)$ score, i.e.,

$$k_0 = \underset{k}{\mathrm{argmin}} \mathrm{AIC_c}(k). \tag{10}$$

Substituting $k_0$ into the weights in Equation (7) to fully determine the elements of the weight matrix $\boldsymbol{W}(u_0, v_0)$, the interpolated value $z_0$ of the variable $Y$ at $(u_0, v_0)$ is consequently obtained by Equation (5). By taking $(u_0, v_0)$ to be each of the candidate interpolation points, say $\{(\widetilde{u}_i, \widetilde{v}_i)\}_{i=1}^m$, we then obtain the interpolated values of $Y$ by

$$z_i = (1, 0, 0)\boldsymbol{Q}(\widetilde{u}_i, \widetilde{v}_i)\boldsymbol{y}, i = 1, 2, \cdots, m, \tag{11}$$

where $\boldsymbol{Q}(\widetilde{u}_i, \widetilde{v}_i)$ is the matrix shown in Equation (6) with $(u_0, v_0)$ replaced by $(\widetilde{u}_i, \widetilde{v}_i)$.

2.1.2. Interpolated-Value-Based Local Spatial Statistics with the Bootstrap Test for Significance of Local Spatial Association

We first lattice under a properly resolution the whole region $D$ as a grid tessellation consisting of, for example, squares with same size. Here we denote by $(\widetilde{u}_i, \widetilde{v}_i)$ the spatial coordinates of the centroid of each grid and by $m$ the number of the grids with their centroids being on the region $D$. With this lattice partition of the region $D$, a spatial proximity matrix, which we denote by $\widetilde{\boldsymbol{W}} = (w_{ij})_{m \times m}$ in order to distinguish it from the previous diagonal weight matrix $\boldsymbol{W}(u_0, v_0)$ in Equation (3), is defined by, for example, the rook or the queen continuity scheme with binary codes 0 and 1.

We then compute the interpolated values of the attribute variable $Y$ at the grid centroids $\{(\widetilde{u}_i, \widetilde{v}_i)\}_{i=1}^m$ according to Equation (11), which we denote by

$$\boldsymbol{z} = (z_1, z_2, \cdots, z_m).$$

Based on $\boldsymbol{z} = (z_1, z_2, \cdots, z_m)$ and $\boldsymbol{W} = (w_{ij})_{m \times m}$, a local spatial statistic can be formulated to measure local spatial association and the related statistical test can further be employed to infer the significance of local spatial association that the statistic measures.

For example, as two of the most commonly used Anselin's LISAs [2], the local Moran's $I_i$ at each $(\widetilde{u}_i, \widetilde{v}_i)$, defined by

$$I_i = \frac{(z_i - \bar{z}) \sum\limits_{j=1}^{m} w_{ij}(z_j - \bar{z})}{\frac{1}{m} \sum\limits_{j=1}^{m} (z_j - \bar{z})^2} \tag{12}$$

with $\bar{z} = \frac{1}{m} \sum\limits_{j=1}^{m} z_j$, and the local Geary's $c_i$ at $(\widetilde{u}_i, \widetilde{v}_i)$, defined by

$$c_i = \frac{\sum\limits_{j=1}^{m} w_{ij}(z_i - z_j)^2}{\frac{1}{m} \sum\limits_{j=1}^{m} (z_j - \bar{z})^2}, \tag{13}$$

measure the local spatial autocorrelation between $z_i$ and its surrounding values, where $w_{ii} = 0 \ (i = 1, 2, \cdots, m)$ are assumed by convention. A positive (negative) value of $I_i$ indicates local positive (negative) spatial autocorrelation or a spatial cluster of similar (dissimilar) values at $(\widetilde{u}_i, \widetilde{v}_i)$. In contrast, a large (small) value of $c_i$ suggests local negative

(positive) spatial autocorrelation. Another popular local spatial statistic is Getis and Ord's $G_i^*$ [8,9], which is defined at $(\widetilde{u}_i, \widetilde{v}_i)$ by

$$G_i^* = \frac{\sum\limits_{j=1}^{m} w_{ij} z_j}{\sum\limits_{j=1}^{m} z_j},\qquad(14)$$

where $z_j > 0$ $(j = 1, 2, \cdots, m)$ are assumed and $w_{jj} > 0$ $(j = 1, 2, \cdots, m)$ are allowed. The local statistic $G_i^*$ is commonly used to identify a spatial cluster of large values (a hot spot) or a spatial cluster of small values (a cold spot) at $(\widetilde{u}_i, \widetilde{v}_i)$ depending upon high or low value of $G_i^*$.

Given a local spatial statistic, the related statistical test is necessary to evaluate significance of the local spatial association that the statistic measures. As mentioned in the introduction section, the bootstrap method is suggested to derive the *p*-value of the related test. Here, we only take the local Moran's $I_i$ as an example to describe the main steps of deriving the *p*-value of testing for positive or negative spatial autocorrelation. The procedures for other commonly used local spatial statistics such as Geary's $c_i$ and Getis and Ord's $G_i^*$ are similar.

The bootstrap procedure for deriving the *p*-value of the local Moran's $I_i$ based test at a given interpolation location $(\widetilde{u}_i, \widetilde{v}_i)$ is as follows.

(i)   Based on the interpolated data $z = (z_1, z_2, \cdots, z_m)$ and the spatial proximity matrix $\widetilde{W} = (w_{ij})_{m \times m}$, compute the observed value of $I_i$ according to Equation (12), which we denote by $I_i^{(0)}$.

(ii)  Draw a bootstrap sample $z^* = (z_1^*, z_2^*, \cdots, z_m^*)$ with replacement from the interpolated data $z = (z_1, z_2, \cdots, z_m)$, on which the bootstrap value of $I_i$, denoted by $I_i^*$, is computed by

$$I_i^* = \frac{\left(z_i^* - \overline{z}^*\right) \sum\limits_{j=1}^{m} w_{ij}\left(z_j^* - \overline{z}^*\right)}{\frac{1}{m} \sum\limits_{j=1}^{m}\left(z_j^* - \overline{z}^*\right)^2},$$

where $\overline{z}^* = \frac{1}{m} \sum\limits_{j=1}^{m} z_j^*$.

(iii) Repeat Step (ii) *B* times and obtain *B* bootstrap values of $I_i$, which we denote by $I_{i(1)}^*, I_{i(2)}^*, \cdots, I_{i(B)}^*$.

(iv)  The *p*-value of testing for positive spatial autocorrelation is

$$p_{i+} = \frac{1}{B}\#\left\{I_{i(j)}^* : I_{i(j)}^* \geq I_i^{(0)}, \; j = 1, 2, \cdots, B\right\},\qquad(15)$$

and the *p*-value of testing for negative spatial autocorrelation is

$$p_{i-} = \frac{1}{B}\#\left\{I_{i(j)}^* : I_{i(j)}^* \leq I_i^{(0)}, \; j = 1, 2, \cdots, B\right\},\qquad(16)$$

where $\#\{A\}$ stands for the number of the elements in the set $A$.

### 2.2. Data Sources for Evaluating the Performance of LGWI and the Bootstrap Test with Interpolated-Value-Based Local Spatial Statistics

In this section, the synthetic spatial data are formulated to evaluate the performance of the proposed LGWI method and the bootstrap test and a real-life example is presented to demonstrate their applicability. Specifically, several data sets from different perspectives are generated by a properly designed experiment to assess the accuracy of the LGWI method and the power of the bootstrap test in identifying spatial association patterns based

on the local spatial statistics constructed by the interpolated values. Furthermore, the proposed LGWI method is compared with the ordinary kriging and IDW interpolation methods to show the superiority of LGWI and the bootstrap test is compared with the conditional permutation test to illustrate their power and computation efficiency. A real-life example based on PM2.5 concentration data is given to demonstrate the applicability of the proposed LGWI method and the bootstrap test with the interpolated-value-based local spatial statistics for identifying significant spatial association patterns.

### 2.2.1. Synthetic Data for Evaluating Accuracy of LGWI and Power of the Test

**(i)   Designed spatial region, sampling points and interpolation points.**

We took the unit square $D = [0, 1] \times [0, 1]$ in a Cartesian coordinate system as the studied spatial region. The following two types of irregularly distributed sampling points of size $n = 200$ were designed.

(a)   Uniformly distributed sampling points on $D$: 200 pairs of random numbers were independently drawn from the uniform distribution $U(0, 1)$ with each pair of the random numbers $(u_i, v_i)$ forming a sampling point on $D$.

(b)   Unevenly distributed sampling points on $D$: 100 pairs of random numbers $\{(u_i, v_i)\}_{i=1}^{100}$ were drawn from the normal distribution $N\left(1/4, \ (9/50)^2\right)$, where only the points in $D$ were retained and the others were discarded until 100 sampling points were obtained. With the same way, the other 100 pairs of random numbers were drawn from $N\left(3/4, \ (1/5)^2\right)$. Because of different means and variances of the two normal distributions, the sampling points form roughly two clusters around the means $1/4$ and $3/4$ respectively and are sparely distributed at the upper-left and lower-right corners.

The interpolation points of the size $m = 400$ were designed as the lattice points on $D$. Specifically, we equally partitioned the unit square $[0, 1] \times [0, 1]$ as 400 small squares by linking the equally spaced points on the two pairs of the parallel sides of the unit square and took the centroid point of each small square as an interpolation point. The coordinates of the interpolation points can be expressed as

$$(\widetilde{u}_i, \widetilde{v}_i) = \left(\frac{1}{20}\mathrm{mod}\left(\frac{i-1}{20}\right) + \frac{1}{40}, \ \frac{1}{20}\mathrm{int}\left(\frac{i-1}{20}\right) + \frac{1}{40}\right), \ i = 1, 2, \cdots, m,$$

where $\mathrm{mod}\,(a/b)$ and $\mathrm{int}(a/b)$ denote the remainder and the integer part of $a$ divided by $b$, respectively, and $m = 400$.

The two types of the sampling points and the interpolation points are depicted in Figure 1, which were taken to be fixed throughout the simulation study.

**(ii)   Model for generating data.**

The model for generating the synthetic spatial data is

$$Y = f(u, v) + \varepsilon, \tag{17}$$

where the model error $\varepsilon$ follows the normal distribution $N(0, \ 0.5^2)$ and the following three underlying spatial processes with different levels of spatial heterogeneity were considered:

(a)   $f_1(u, v) = 2(u + v);$
(b)   $f_2(u, v) = 4\sin(\pi u);$
(c)   $f_3(u, v) = 1 + 600(uv)^5(1 - u)(1 - v).$

The true surfaces of the above spatial processes are shown in the first column of both Figures 3 and 6.

Given each of the above spatial processes $f_j(u,v)$ $(j = 1,2,3)$, the observations $\{y_i\}_{i=1}^n$ of $Y$ with $f(u,v) = f_j(u,v)$ in Equation (17) were generated at all of the $n = 200$ sampling points $\{(u_i, v_i)\}_{i=1}^n$ by

$$y_i = f_j(u_i, v_i) + \varepsilon_i, \quad i = 1,2,\cdots,n, \tag{18}$$

where $\{\varepsilon_i\}_{i=1}^n$ are the random numbers independently drawn from $N(0,\ 0.5^2)$.

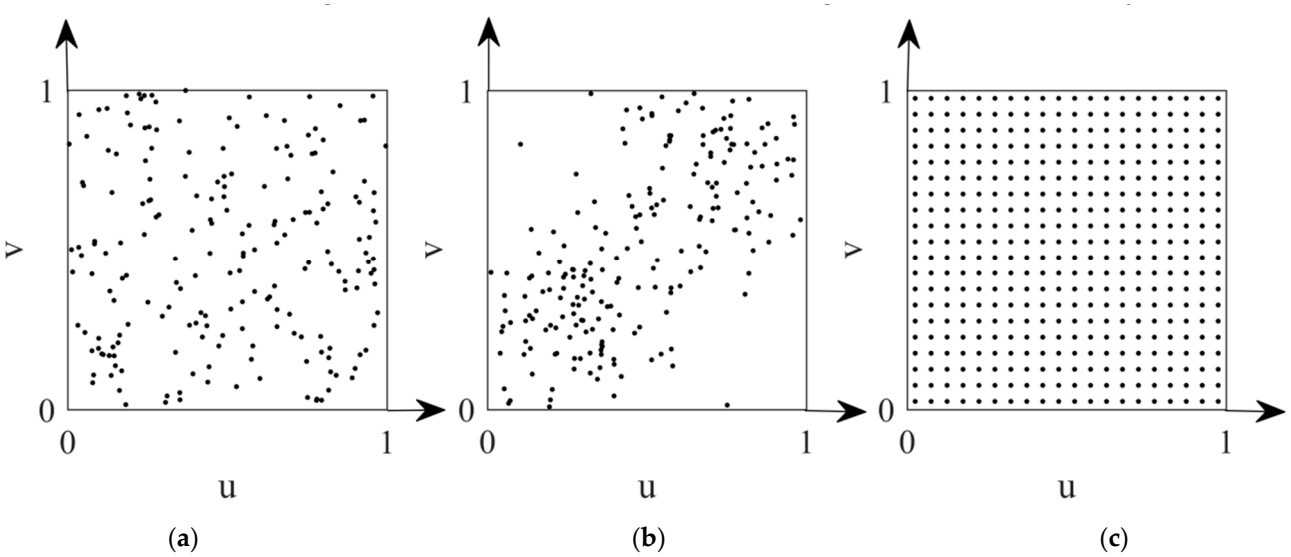

**Figure 1.** Two types of sampling points and the lattice interpolation points. (**a**) uniformly distributed sampling points; (**b**) unevenly distributed sampling points; (**c**) lattice interpolation points.

**(iii)  Indices for measuring accuracy of the interpolation methods and power of the tests.**

Based on the data $\{y_i\}_{i=1}^n$ generated by each of the foregoing spatial processes, the proposed LGWI procedure as well as the kriging and IDW methods for comparison were used to derive the interpolated values $\{z_i\}_{i=1}^m$ at the interpolation points $\{(\widetilde{u}_i, \widetilde{v}_i)\}_{i=1}^m$. In order to alleviate sampling error, $N$ replications in each experimental setting were run, where the model errors $\{\varepsilon_i\}_{i=1}^m$ were re-drawn in each replication. For each interpolation method, let $z_{1(j)}(\widetilde{u}_i, \widetilde{v}_i), z_{2(j)}(\widetilde{u}_i, \widetilde{v}_i), \cdots, z_{m(j)}(\widetilde{u}_i, \widetilde{v}_i)$ be the interpolated values at $(\widetilde{u}_i, \widetilde{v}_i)$ in the $j$-th replication, we took their average value

$$\overline{z}(\widetilde{u}_i, \widetilde{v}_i) = \frac{1}{N}\sum_{j=1}^N z_{i(j)}(\widetilde{u}_i, \widetilde{v}_i) \tag{19}$$

for each of $i = 1,2,\cdots,m$ as the final interpolated value of the underlying spatial process at $(\widetilde{u}_i, \widetilde{v}_i)$. As well known, the mean square error at each $(\widetilde{u}_i, \widetilde{v}_i)$ is

$$\text{MSE}(\widetilde{u}_i, \widetilde{v}_i) = \frac{1}{N}\sum_{j=1}^N \left(z_{i(j)}(\widetilde{u}_i, \widetilde{v}_i) - f(\widetilde{u}_i, \widetilde{v}_i)\right)^2, \ i = 1,2,\cdots,m.$$

We use in this article the square root of the averaged mean square errors (RAMSE) over all of the interpolation points, showing

$$\text{RAMSE} = \sqrt{\frac{1}{m}\sum_{i=1}^m \text{MSE}(\widetilde{u}_i, \widetilde{v}_i)}, \tag{20}$$

as an index to measure the global accuracy of each interpolation method for the underlying spatial process $f(u,v)$.

Given a significance level $\alpha$, the rate of rejecting the null hypothesis at an interpolation point $(\widetilde{u}_i, \widetilde{v}_i)$ in the $N$ experiment replications, which we denote by $r_N(\widetilde{u}_i, \widetilde{v}_i)$, is employed

to assess the performance of the bootstrap and conditional permutation tests. As well known in statistics, $r_N(\tilde{u}_i, \tilde{v}_i)$ measures the type-I error of a test when the null hypothesis at $(\tilde{u}_i, \tilde{v}_i)$ is true and the power of the test when the alternative hypothesis is true. If the test is well formulated, $r_N(\tilde{u}_i, \tilde{v}_i)$ should be close to the significance level $\alpha$ when no spatial association exists; it should be large enough when spatial association does exist at $(\tilde{u}_i, \tilde{v}_i)$. Therefore, the larger the rejection rate $r_N(\tilde{u}_i, \tilde{v}_i)$ is, the more powerful the test is in identifying local spatial association at $(\tilde{u}_i, \tilde{v}_i)$.

2.2.2. Real-Life Data for Demonstrating Applicability of LGWI and the Test Methods

To demonstrate the applicability of the proposed LGWI method and the suggested bootstrap test for identifying significant spatial association patterns, the PM2.5 concentration data in Guangdong province, China, are chosen to achieve the task. The reasons for choosing such data are two-fold. On one hand, PM2.5 concentrations are the typical geostatistical data that this article mainly focuses on. On the other hand, the air quality monitoring stations where the data are collected in Guangdong province, as shown in the left panel of Figure 2, are highly clustered and unevenly distributed over space, which is just the challenge for the LGWI method to overcome.

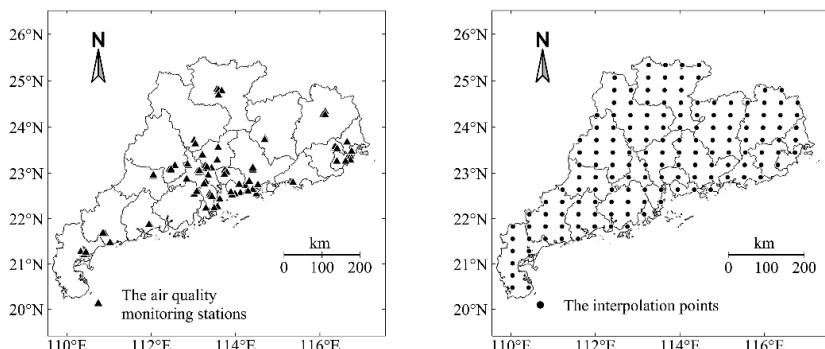

**Figure 2.** Spatial distributions of the air quality monitoring stations and the interpolation points in Guangdong province.

**(i)    Original data with preprocessing.**

The original data, which are available at http://air.cnemc.cn:18007/ (accessed on 20 August 2020), include the hourly PM2.5 concentrations collected from 101 air quality monitoring stations in Guangdong province, China, in 2019. The longitude and latitude of each monitoring station are also attached. In the original data, there are a few missing data. Firstly, we replenished each missing datum with the mean of its preceding and subsequent data, which is reasonable because the time duration is only 1 h between two adjacent data. Furthermore, since the Cartesian coordinates of the spatial locations for interpolation are needed in the proposed LGWI method, we therefore employed the simplified latitude planar projection [32] to transform the longitude and latitude to the Cartesian coordinates, where the horizontal axis is the equator of the Earth and the vertical axis is the meridian crossing the origin of the Xi'an 80 coordinate system of China with the longitude being 108.92°. Specifically, let $(\lambda, \varphi)$ be the longitude and latitude of a location on the Earth, its Cartesian coordinates $(u, v)$ transformed by the simplified planar projection are

$$\begin{cases} u = R\dfrac{\pi}{180}(\lambda - 108.92)\cos\left(\dfrac{\pi}{180}\varphi\right); \\ v = R\dfrac{\pi}{180}\varphi, \end{cases}$$

where $R = 6367.554$ km is the radius of the Earth. We then latticed the whole region of the province as $m = 152$ grids and took the centroids of the grids as the interpolation points. With the foregoing notations, we denoted the Cartesian coordinates of the sampling

points (i.e., the spatial locations at which the monitoring stations locate) by $\{(u_i, v_i)\}_{i=1}^{n}$ with $n = 101$ and those of the interpolation points by $\{(\widetilde{u}_i, \widetilde{v}_i)\}_{i=1}^{m}$ with $m = 152$. Figure 2 shows the sampling points and the interpolation points on the map of Guangdong province, from which we can observe that the sampling points are highly clustered and unevenly distributed over the province.

**(ii)** **Data sets formulated to explore seasonal local spatial association patterns of PM2.5 concentration.**

The typical climatic characteristic in Guangdong province, China, is the subtropical monsoon climate, which may make air pollution show different patterns in different seasons. Therefore, it is of interest to identify the patterns of local spatial association of PM2.5 concentration in different seasons especially the areas where the level of PM2.5 concentration is significantly high or low. This task can be well achieved by simultaneously using the interpolated-value-based local Moran's $I_i$ and local Getis and Ord's $G_i^*$. According to aerography, January, April, July and October are four typical months standing for winter, spring, summer and autumn seasons in the northern hemisphere, respectively. We therefore took the preprocessed data in these four months of each monitoring station to conduct the case study. Specifically, the hourly PM2.5 concentrations in each of the four months were averaged at each monitoring station as the monthly PM2.5 concentrations at that station, resulting in the four monthly average data sets with the size of $n = 101$, which will be used to detect seasonal local spatial association patterns of PM2.5 concentration in Guangdong province.

## 3. Results with Comments

*3.1. Simulation Results of Evaluating Accuracy of the LGWI Method and Power of the Test*

### 3.1.1. Accuracy of LGWI

Here, the number of the experiment replications was set to be $N = 500$. In addition to the proposed LGWI method, the ordinary kriging interpolation with a spherical semi-variogram and an exponential semi-variogram and the IDW interpolation with the values of the power parameter being respectively 1 and 2 were also conducted for each experimental setting under the same synthetic data sets. Table 1 reports the resulting values of RAMSE for the three interpolation methods in each experimental setting. The final interpolated surfaces characterized by the values $\{\bar{z}(\widetilde{u}_i, \widetilde{v}_i)\}_{i=1}^{m}$ defined in Equation (19) for each of the three interpolation methods under the two sampling schemes are depicted in Figures 3–5 with the true surfaces of the three underlying spatial processes attached in the first column of Figure 3 for comparison.

**Table 1.** Values of RAMSE for the three interpolation methods under 500 experimental replications.

| Interpolated Method | Spatial Process | Uniformly Sampling Scheme | Unevenly Sampling Scheme |
|---|---|---|---|
| LGWI | $f_1(u,v)$ | 0.0762 | 0.0997 |
| | $f_2(u, v)$ | 0.2018 | 0.2797 |
| | $f_3(u,v)$ | 0.2302 | 0.2450 |
| Kriging (spherical semi-variogram) | $f_1(u,v)$ | 0.1659 | 0.1830 |
| | $f_2(u, v)$ | 0.3162 | 0.3879 |
| | $f_3(u,v)$ | 0.2074 | 0.2144 |
| Kriging (exponential semi-variogram) | $f_1(u,v)$ | 0.1607 | 0.1757 |
| | $f_2(u, v)$ | 0.2891 | 0.3840 |
| | $f_3(u,v)$ | 0.2089 | 0.2146 |
| IDW (power parameter = 1) | $f_1(u,v)$ | 0.4824 | 0.4004 |
| | $f_2(u, v)$ | 0.9300 | 1.0080 |
| | $f_3(u,v)$ | 0.4194 | 0.4067 |
| IDW (power parameter = 2) | $f_1(u,v)$ | 0.2591 | 0.2186 |
| | $f_2(u, v)$ | 0.5029 | 0.7242 |
| | $f_3(u,v)$ | 0.2803 | 0.3017 |

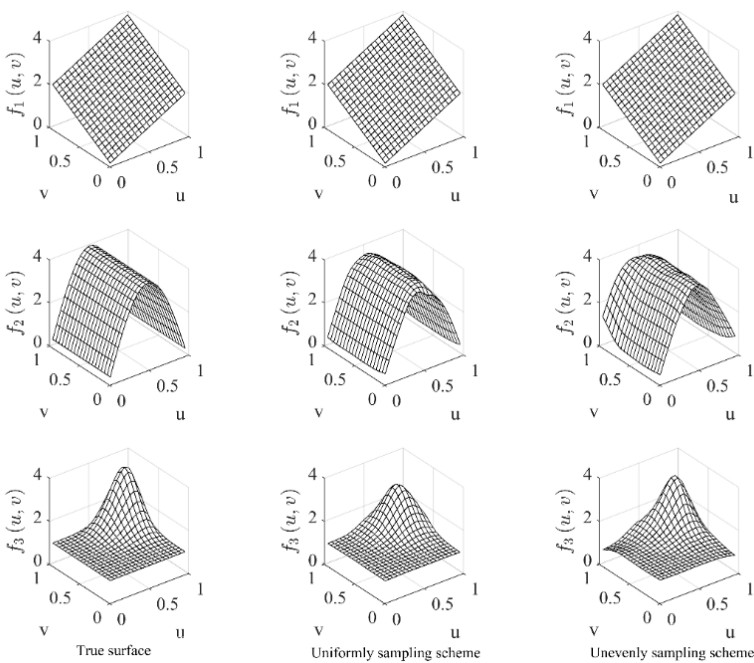

**Figure 3.** True surfaces (the first column) and the interpolated surfaces (the second and the third columns) of the three spatial processes by the LGWI method.

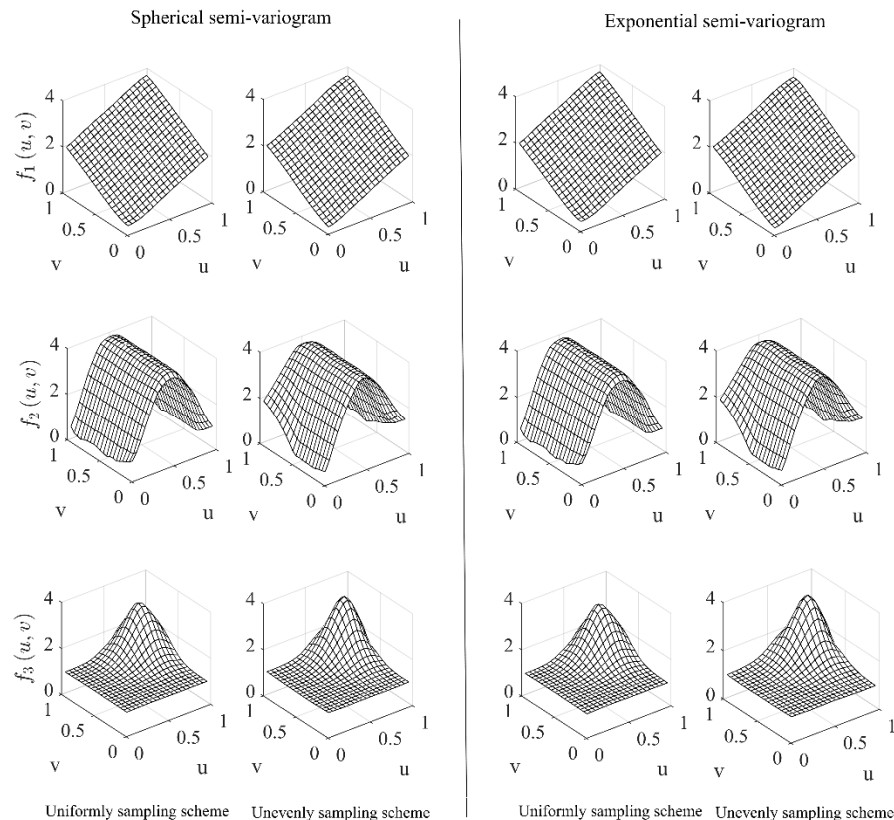

**Figure 4.** Interpolated surfaces of the three spatial processes by the ordinary kriging method.

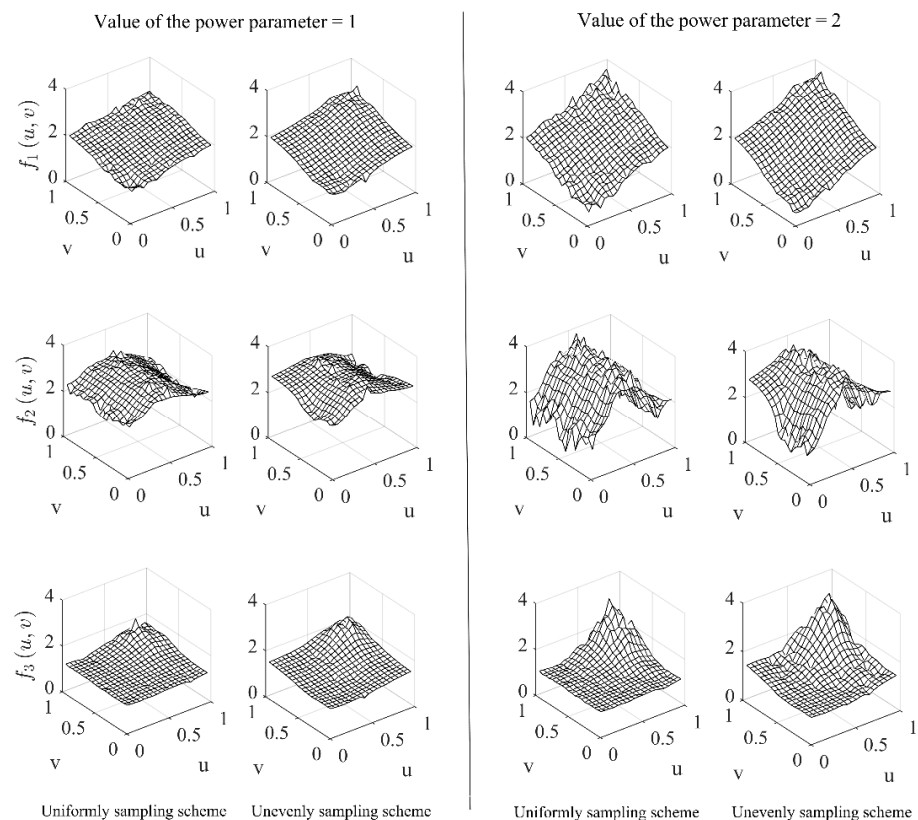

**Figure 5.** Interpolated surfaces of the three spatial processes by the IDW interpolation method.

Focusing on the accuracy of LGWI first, we observe from Table 1 that the LGWI method performs especially well for both sampling schemes when the underlying spatial process is a linear function of spatial coordinates. Although the accuracy decreases with spatial heterogeneity of the underlying process increasing and with the sampling points becoming unevenly distributed over space, the values of RAMSE are very small comparing with the range of about the interval [0, 4] for each of the spatial processes. Ordinary kriging interpolation seems somewhat robust to spatial heterogeneity of the underlying process and to the chosen semi-variograms. However, except for the comparable accuracy to that of LGWI for $f_3(u, v)$, the kriging method yields significantly larger values of RAMSE for the other two spatial processes even for the simplest linear function. The IDW interpolation method seems quite sensitive to the value of power parameter and performs worst among the three interpolation methods. It yields much larger values of RAMSE than the proposed LGWI and the ordinary kriging methods. In summary, LGWI performs best in terms of RAMSE.

Figures 3–5 show more detailed information about the interpolated surfaces. It is observed that, under both sampling schemes, both LGWI and kriging retrieve true surfaces of the underlying spatial processes quite well. However, the interpolated surfaces of $f_2(u, v)$ produced by kriging show more serious distortion on the boundary area especially under the unevenly sampling scheme. In contrast, IDW interpolation yields very rough interpolated surfaces even for the linear function $f_1(u, v)$. More seriously, the basic spatial pattern of $f_2(u, v)$ is not well retrieved.

### 3.1.2. Power of the Bootstrap Test with the Interpolated-Value-Based Local Statistics in Identifying Local Spatial Association

The foregoing simulation results demonstrate that all of the interpolated surfaces by LGWI can well retain their respective spatial patterns of the underlying spatial processes, which makes it reliable to use the LGWI interpolated-value-based local spatial statistics

to detect local spatial association of the underlying processes. Here the interpolated-value-based local spatial statistics $I_i$ and $G_i^*$ were exemplified to evaluate the power of the bootstrap test in identifying local spatial association of the underlying processes, in which the queen spatial scheme was chosen to formulate the binary spatial proximity matrix $\widetilde{W} = (w_{ij})_{m \times m}$ for the grids of the interpolation spatial layout shown in Figure 1c. Namely, $w_{ij} = 1$ if grids $i$ and $j$ have a common vertex and $w_{ij} = 0$ if otherwise; $w_{ii} = 0$ $(i = 1, 2, \cdots, m)$ in $I_i$ and $w_{ii} = 1$ $(i = 1, 2, \cdots, m)$ in $G_i^*$ are assumed. The same data sets in the foregoing $N$ experiment replications for obtaining the interpolated values of each spatial process were used to compute the rejection rate at each interpolated point.

What we focus here is to identify local positive autocorrelation using $I_i$ and hot spots using $G_i^*$. Specifically, in each replication, we first obtained the interpolated values $\{z_i\}_{i=1}^m$ at the interpolation points $\{(\widetilde{u}_i, \widetilde{v}_i)\}_{i=1}^m$ with $m = 400$, on which the values of $I_i$ and $G_i^*$ were computed at each interpolation point $(\widetilde{u}_i, \widetilde{v}_i)$. Then, the bootstrap procedure was used to derive the $p$-values of the $I_i$ and $G_i^*$ based tests at each $(\widetilde{u}_i, \widetilde{v}_i)$, where the number of the bootstrap sampling was set to be $B = 500$ and the $p$-values of $p_{i+}$ for testing positive spatial association were computed for $I_i$ and $G_i^*$, respectively. Given a significance level $\alpha$, if $p_{i+} < \alpha$, we rejected the null hypothesis that there is no positive spatial association at $(\widetilde{u}_i, \widetilde{v}_i)$. After the $N = 500$ replications were run, the rejection rate $r_N(\widetilde{u}_i, \widetilde{v}_i)$ at each $(\widetilde{u}_i, \widetilde{v}_i)$ was obtained.

The rejection rates $\{r_N(\widetilde{u}_i, \widetilde{v}_i)\}_{i=1}^m$ with $\alpha = 0.05$ under the unevenly sampling scheme for each of the spatial processes are shown via the heat maps in Figure 6, where the true surfaces of the three spatial processes are also attached for the purpose of comparison. For the uniformly sampling scheme, the corresponding heat maps show the similar but more evident patterns for each of the spatial processes because of the more accurate interpolated surfaces shown in the second column of Figure 3. Therefore, the heat maps under the uniformly sampling scheme are omitted here to save space.

Comparing the spatial patterns of the underlying spatial processes with the corresponding heat maps of the rejection rates, we can clearly observe that both Moran's $I_i$ based and Getis and Ord's $G_i^*$ based tests can powerfully identify the grids where local positive spatial association exists. Specifically, $I_i$ based test can correctly identify the locations where the similar (either low or high) values cluster with their rejection rates are all close to 1, meaning that the locations where positive spatial autocorrelation exists can be almost surely identified. $G_i^*$ based test is also powerful in identifying hot spots or locations where larger values cluster. The rejection rates where positive spatial association are not significant are all less than or approximately equal to the significance level of $\alpha = 0.05$, indicating that the bootstrap test has a valid nominal probability. Although the inaccurate interpolated values in the area where the sampling points are very sparse have somewhat adverse effect on the test power, the basic spatial association patterns of the underlying processes are still well uncovered by the bootstrap test.

Furthermore, for the purpose of comparison, the conditional permutation test [2] was conducted based on the same interpolated values by LGWI under the unevenly sampling scheme and the same local spatial statistics for identifying positive spatial association. The conditional permutation test is similar to the forgoing bootstrap test in the principle and procedures except that the permutation samples are obtained by fixing the observation at each focal sampling point and permutating the other observations among the remainder sampling points, which is, as mentioned in [12], slightly more complicated than bootstrap sampling where the bootstrap samples are drawn with replacement. We obtain the almost same heat maps of the reject rates as those shown in Figure 6a,b, which we omit here to save place. Such results are in accord with the finding in [12] for spatiotemporal data. In view of the computation efficiency, the two test methods are almost similar in our simulation which took about 6 s on our common personal computer for each experiment replication with the sample size $m = 400$. However, in the case of large sample size of more than 2000, Yan et al. [12] found that the bootstrap test is more efficient than the permutation test. The above comparison demonstrates that the conditional permutation test could be an

alternative choice for identifying significant spatial association patterns of an underlying spatial process.

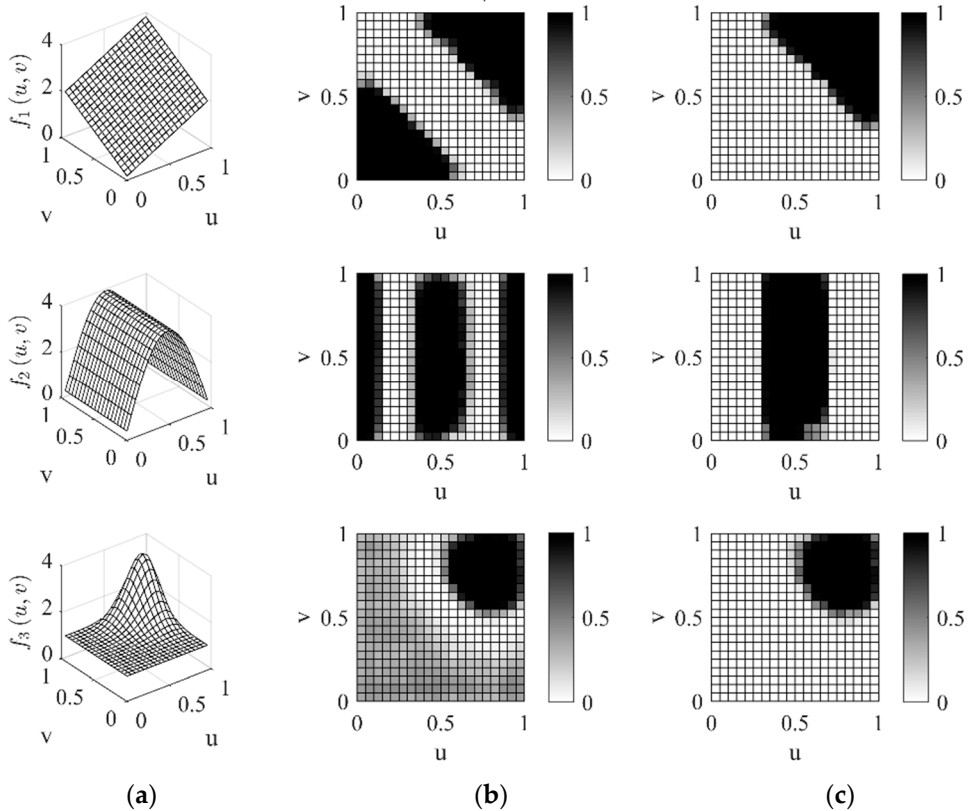

**Figure 6.** True surfaces of the three spatial processes and the heat maps of the rejection rates with the significance level $\alpha = 0.05$ for testing local positive spatial association based on $I_i$ and $G_i^*$ under the unevenly sampling scheme. Column (**a**) true surfaces; column (**b**) heat maps of the rejection rates for $I_i$; column (**c**) heat maps of the rejection rates for $G_i^*$.

### 3.2. Seasonal Local Spatial Association Patterns of PM2.5 Concentration in Guangdong Province

As mentioned in Section 2.2.2 (ii), what we are interested in is to identify the areas where PM2.5 concentration is significantly high and the areas where it is significantly low, which can be achieved, as has shown in the simulation study, by simultaneously using $I_i$ and $G_i^*$ to explore local positive spatial association at each interpolation point. With the $p$-values of the $I_i$ and $G_i^*$ based tests for the significance of local positive spatial association, the interpolation points where both $p$-values of the $I_i$ and $G_i^*$ based tests are less than a given significance level, for example, $\alpha = 0.05$ indicate the areas where PM2.5 concentration is high, while the interpolation points where only the $p$-values of the $I_i$ based test are less than $\alpha$ imply the areas where PM2.5 concentration is low.

Along the above line of reasoning, for each of the four data sets formulated in Section 2.2.2, the LGWI procedure was used to obtain the interpolated PM2.5 concentrations at the interpolation points $\{(\tilde{u}_i, \tilde{v}_i)\}_{i=1}^m$ with $m = 152$, on which both $I_i$ and $G_i^*$ were used to locally measure spatial association among the PM2.5 concentrations. In the calculation of the local spatial statistics, the queen spatial scheme among the grids was once again taken to specify the spatial proximity matrix $\widetilde{W} = (w_{ij})_{m \times m}$. In the bootstrap test, $B = 500$ bootstrap samples were drawn from the interpolated values and the $p$-values for testing positive spatial association were computed for both $I_i$ and $G_i^*$ based tests. Figure 7 shows the heat maps of the interpolated values of PM2.5 concentration on the grids on which the corresponding interpolated values rounded to integers were overlapped and the choropleth maps of the $p$-values of the $I_i$ and $G_i^*$ based tests for the four months.

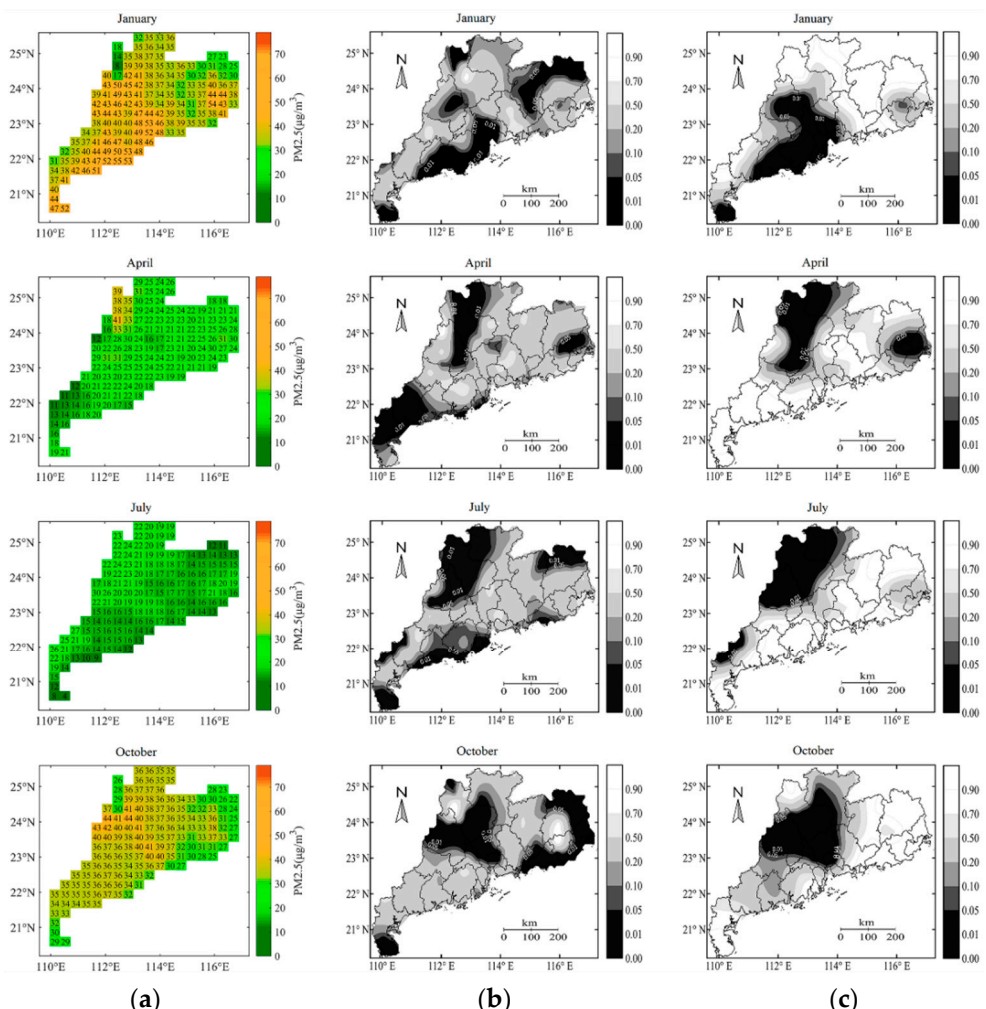

**Figure 7.** Heat maps of the interpolated surfaces of PM2.5 concentration and choropleth maps of $p$-values of the $I_i$ and $G_i^*$ based tests for identifying positive spatial association in the four months. Column (**a**) interpolated surface; column (**b**) $p$-values of $I_i$ based test; column (**c**) $p$-values of $G_i^*$ based test.

It can be observed from the heat maps in the first column of Figure 7 that, as a whole, the PM2.5 concentration is highest in January, the typical month in winter, while it is lowest in July, the typical month in summer. In addition, the heat maps in the four months all show obvious spatial heterogeneity over the province. The choropleth maps of the $p$-values of $I_i$ and $G_i^*$ based tests attached in the second and third columns of Figure 7 can be jointly used, as aforementioned, to uncover the areas where the PM2.5 concentration is heavy or light in each specific month. According to the choropleth maps of the $p$-values, it can be concluded that, under the significance level of $\alpha = 0.05$, the heavily polluted areas are in the southwest part of the province while the lightly polluted areas locate at the north part in January; in April, however, the situation is opposite. In July, the areas with high PM2.5 concentration are along the northwest boundary of the province and those with low PM2.5 concentration are mainly along the costal line with an additional area being in the northeast. In October, besides the lightly polluted area in the southernmost part, both heavily and lightly polluted areas are in the north part with the heavily polluted area being in the west and the lightly polluted area being along the eastern boundary. In summary, both heavily polluted areas and lightly polluted ones by PM2.5 pollutant change over the province from season to season, showing different spatial association patterns in each season.

## 4. Conclusions and Discussion

This paper proposes the LGWI approach for geostatistical data to deal with the problem that the data are only available at unevenly distributed locations over the studied region, making it troublesome to construct a local spatial statistic for the exploration of local patterns of spatial association over the whole studied region. This method can implement interpolation on a lattice spatial tessellation on which a local spatial statistic can be well defined. Furthermore, the bootstrap test is suggested to identify significant local spatial association patterns of an underlying spatial process based on interpolated-value-based local spatial statistics. The simulation study has shown that the proposed interpolation method can adequately retrieve the true patterns of underlying processes with different levels of spatial heterogeneity and the bootstrap test is powerful in identifying significant local patterns of spatial association. The case study of PM2.5 concentration has demonstrated that the interpolation and the test methods perform well in comprehensively identifying spatial association patterns of real-life geostatistical data with their sampling points unevenly distributed and highly clustered over space.

The performance comparison made in the simulation study shows that the proposed LGWI interpolation is more flexible than the existing commonly used kriging and IDW interpolation methods and yields more accurate interpolated values for an underlying spatial process. The comparison between the bootstrap test and the conditional permutation test demonstrates the latter could also be taken as an alternative method for identifying significant local spatial association patterns especially when the sample size is moderate or small. Although the proposed LGWI interpolation method is motived by overcoming the challenge that the unevenly distributed sampling points make it difficult to use local spatial statistics for the exploration of local spatial association among geostatistical data, it contributes a new method to the toolbox of interpolation methodologies. In addition, although the proposed interpolation and test methods were evaluated for their performance and applicability by the local spatial statistics $I_i$ and $G_i^*$ and PM2.5 concentration data in this article, they are of generality to be used for other local spatial statistic and any a geostatistical data set.

Nevertheless, it should be noted that, like any a local spatial statistic based test, the interpolated-value-based test involves the multiple comparison issue because the test should be in general conducted over all of the interpolation points in order to comprehensively uncover the interested local spatial association patterns of the underlying process. As pointed in [2], the Bonferroni and the Sidák methods for dealing with the multiple comparison issue are usually very conservative. Fortunately, the false discovery rate (FDR) criterion proposed by Benjamini and Hochberg [33] has been widely applied to a variety of fields for handling the multiple comparison issue. This criterion has also been used to deal with the multiple comparison issue in local spatial statistic based tests and the results have shown that the FDR criterion is more powerful than the Bonferroni and the Sidák methods by de Castro and Singer [34]. Therefore, it is expected that the results for testing significant spatial association patterns in this paper would be more convincing if some well performed procedure for dealing with the multiple comparison issue was considered in the interpolated-value-based bootstrap test.

As a future research direction, it seems possible to extend LGWI and the related testing procedure to identify spatiotemporal association for geostatistical spatiotemporal data. The interpolation at a lattice spatiotemporal tessellation can be implemented by assuming the nonparametric regression model $y = f(u, v, t) + \varepsilon$ and obtaining the predictor at each interpolation point by the local-linear geographically weighted least-squares procedure, where the spatiotemporal distance like that in the geographically and temporally weighted regression technique [35] could be used to generate the spatiotemporal weights. Then, based on the latticed spatiotemporal interpolation points and the interpolated values of $Y$, some local spatiotemporal statistics [10–12,36] can be constructed and the bootstrap test developed in [12] or the randomized permutation procedure could be used to identify significance of local spatiotemporal association of the underlying spatiotemporal process.

In view of the wide application backgrounds of identifying local association among geo-statistical spatiotemporal data, the extension of the interpolation and the test methods to geostatistical spatiotemporal data deserves to be studied in the future.

**Author Contributions:** Conceptualization and methodology, C.-L.M.; software and visualization, F.-J.W. and Z.Z.; writing—original draft preparation, C.-L.M.; writing—review and editing, C.-L.M., F.-J.W., Z.Z. and Q.-X.X. All authors have read and agreed to the published version of the manuscript.

**Funding:** This research was funded by the National Natural Science Foundation of China (grant numbers 11871056 and 11271420).

**Institutional Review Board Statement:** Not applicable.

**Informed Consent Statement:** Not applicable.

**Data Availability Statement:** The original PM2.5 concentration data in Guangdong province, China, are freely available at http://air.cnemc.cn:100871/ (accessed on 20 August 2022).

**Acknowledgments:** The authors sincerely thank the reviewers for their valuable and constructive comments and suggestions which led to significant improvement on the manuscript.

**Conflicts of Interest:** The authors declare no conflict of interest related to this work.

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
