# Peer review of "Testing for Local Spatial Association Based on Geographically Weighted Interpolation of Geostatistical Data with Application to PM2.5 Concentration Analysis"

_sustainability, doi:10.3390/su142114646_

Round 1

Reviewer 1 Report

A local-linear geographically weighted interpolation method is proposed to obtain the predictors of the underlying spatial process on a lattice spatial tessellation. The bootstrap test is employed to identify the locations where local spatial association is significant. Some specific suggestions and comments are as follows.

(1) If this paper is a methodological paper, it should use different types of data to validate the method, whereas, only one kind of data is used. If this paper is to discuss the PM2.5 concentration, the research on PM2.5 concentration should be systematically reviewed in the introduction, whereas, there is almost no literature on PM2.5.

(2) Figure 2: Some cartographic elements, such as legend, scale, and compass, should be added.

(3) Can the uniform sampling scheme and the clustered sampling scheme be described quantitatively? Otherwise, it is difficult to describe the two schemes.

(4) Figure 5. Column (a): Because the grid size is too large, the spatial resolution is very low and the generated raster map has a very poor effect.

(5) Many interpolation approaches have been proposed, such as kriging interpolation and IDW (Inverse Distance Weighted). As a methodological paper, it is necessary to carry out a comparison of interpolation approaches. 

(6) The discussion of the paper should be strengthened. Moreover, an academic paper should have a conclusion section.

Reviewer 2 Report

This paper discusses the problem of detecting local spatial association for geo-referenced data. The main challenge of this problem is that geostatistical data such as meteorological data and air pollution data are generally collected from meteorological or monitoring stations which are usually sparsely located or highly clustered over the region, and that a local spatial statistic formulated at an isolate sampling point may be ineffective because of its distant neighbors; or the statistic is undefinable in the sub-regions where no observations are available, which limits the comprehensive exploration of local spatial association over the whole region. To overcome the challenge, the authors propose a method combined by a local linear geographically weighted interpolation and bootstrap tests. The method is evaluated on a synthetic and a real-world dataset. A case study is conducted to show that the method can detect local spatial association.

To improve the paper, there are several issues that can be resolved.
1. The motivation of the study is not clear. In other words, why do we need to detect local spatial association patterns? The authors present a case study on detecting PM2.5 concentration in this paper, but do not say why they need it. In addition, are there any other applications of this study?
2. The problem is not well-defined. What are the input and output of the problem?
3. The related work is not well-surveyed. Most of the related work discussed in the paper is old. There are many works on spatial clustering and spatial hotspot that are also related to this problem, including works on spatial lattice data and statistical significance. The following survey lists many methods on this topic.

Xie, Yiqun, Shashi Shekhar, and Yan Li. "Statistically-robust clustering techniques for mapping spatial hotspots: A survey." ACM Computing Surveys (CSUR) 55, no. 2 (2022): 1-38.

4. The experiments using synthetic and real-world datasets are not enough to evaluate the method. There are two parts in the proposed method: an interpolation method and a bootstrap test, but the experiments do not compare them with the related work.

Round 2

Reviewer 1 Report

The authors have revised the paper according to my suggestions, and I would like to see the paper published in Sustainability.

Reviewer 2 Report

I don't have any other comments.